# Modulation of Distribution and Diffusion through the Lipophilic Membrane with Cyclodextrins Exemplified by a Model Pyridinecarboxamide Derivative

**DOI:** 10.3390/pharmaceutics15051531

**Published:** 2023-05-18

**Authors:** Tatyana Volkova, Olga Simonova, German Perlovich

**Affiliations:** G.A. Krestov Institute of Solution Chemistry RAS, 153045 Ivanovo, Russia; vtv@isc-ras.ru (T.V.); ors@isc-ras.ru (O.S.)

**Keywords:** solubility, distribution, membrane permeability, PermeaPad barrier, cyclodextrins

## Abstract

The main aims of the study were to disclose the influence of the structure on the solubility, distribution and permeability of the parent substances, iproniazid (IPN), isoniazid (INZ) and isonicotinamide (iNCT), at 310.2 K and to evaluate how the presence of cyclodextrins (2-hydroxypropyl-β-cyclodextrin (HP-β-CD) and methylated β-cyclodextrin (M-β-CD)) affects the distribution behavior and diffusion properties of a model pyridinecarboxamide derivative, iproniazid (IPN). The following order of decreasing the distribution and permeability coefficients was estimated: IPN > INZ > iNAM. A slight reduction of the distribution coefficients in the 1-octanol/buffer pH 7.4 and n-hexane/buffer pH 7.4 systems (more pronounced in the first system) was revealed. The extremely weak IPN/cyclodextrins complexes were estimated from the distribution experiments: K_C_(IPN/HP-β-CD) > K_C_(IPN/M-β-CD). The permeability coefficients of IPN through the lipophilic membrane—the PermeaPad barrier—were also measured with and without cyclodextrins in buffer solution. Permeability of iproniazid was increased in the presence of M-β-CD and reduced by HP-β-CD.

## 1. Introduction

Pyridine (a six membered heterocyclic ring, comprising of five carbon atoms and one nitrogen atom) (Figure 1a) is an important pharmacophore and an exceptional heterocyclic system in the field of drug discovery [1]. In combination with other moieties (for example, carboxamide one (Figure 1b)), it serves as a base for the development of new pharmaceutical entities with a large variety of biological and therapeutic applications. For example, the pyridinecarboxamide scaffold is promising in drug design. It is very common in drugs and these drugs can exhibit different biological activity and physicochemical properties depending on the presence of substituents and additional functional groups. Synthesis of pyridinecarboxamide derivatives is an ongoing process. Ammar et al. [2] synthesized a series of pyridinecarboxamide-containing naproxen derivatives through different routes and proposed the derivatization of the well-established drug structures as an approach to modulate toxicity and other possible unpleasant side effects. A series of novel hydrazine derivatives containing the pyridine amide moiety were designed [3] and shown to possess an insecticidal activity. Ali et al. [4] introduced phenol or catechol functional groups to obtain new pyridinecarboxamide containing compounds which demonstrated good bioavailability, water solubility and other drug-like parameters in pharmacokinetic studies, as well as lower toxicity and high activity in a number of biological studies.

In addition to the synthesis of new entities, investigations on existing drugs based on the pyridinecarboxamide moiety are also of interest. Studying the properties of current compounds has pharmacological significance, on the one hand, due to the possibilities of selecting the directions for the synthesis of the structurally-related substances with the desired physicochemical and biological parameters, and, on the other, because this knowledge facilitates the design of bioavailable drug formulations by different techniques (solid dispersions with different excipients, pharmaceutical cocrystals, etc.). In the framework of QSPR (Quantitative Structure–Property Relationship) and QSAR (Quantitative Structure–Activity Relationship) [5] approaches, development of the correct schemes for the targeted synthesis of the drug compounds can be considered as an advantageous tool to minimize unwanted effects [6].

Iproniazid (IPN), isoniazid (INZ) and isonicotinamide (iNCT) (Figure 2) were used in the present study as the structurally related pyridinecarboxamide-containing substances. Isoniazid is one of the most effective antitubercular drugs with a specific high activity against Mycobacterium tuberculosis [7], whereas iproniazid (a non-selective, irreversible monoamine oxidase inhibitor of the hydrazine class) is a xenobiotic that was originally designed to treat tuberculosis, but was later most prominently used as an antidepressant drug [8]. Iproniazid has been replaced in antidepressant therapy by the other members of the monoamine-oxidase inhibitor series and the tricyclic antidepressant drugs, which have less severe side effects. Isonicotinamide has been intensively used in pharmaceuticals due to its anti-tubercular, anti-pyretic, fibrinolytic and anti-bacterial activity [9] and also serves as an advantageous coformer in pharmaceutical cocrystal technology [10].

Design of an appropriate delivery system for a specific drug compound implies the investigation of physicochemical properties relevant for the pharmaceutical industry and selection of suitable components to obtain the most appropriate formulation. Since in order to reach a biologic receptor any drug should dissolve in the physiological fluids, distribute in the body tissues, and penetrate through the biological membranes, the properties of solubility, partition and permeability (so-called transport properties) are of a paramount importance for pharmaceutical chemists and have become an essential part of investigations. The knowledge of these properties within a family of closely related compounds can help to establish the prognostic models for the structure–property correlations with the aim of synthesizing drug substances with the improved properties [11,12].

Cyclodextrins are common pharmaceutical excipients often used to modulate the solubility and permeability in drug formulations. Some of the additional advantages of cyclodextrins include their potential to improve a drug’s stability, safety and organoleptic properties [13,14,15], as well as reduction of toxic action [16,17] and side effects [18]. Many chemically modified β-cyclodextrins, such as hydroxypropyl- or methyl-substituted ones, are successfully used in pharmaceutical products [19]. Cyclodextrins impact the properties of drugs (solubility, distribution, permeability) in different modes, which was demonstrated in the literature [20,21,22,23], including some studies carried out by our scientific team [11,24,25]. This comes from the fact that among the many factors which can influence the character of drug–cyclodextrin interaction, the most important one is the nature of the cyclodextrin used [26,27]. To the best of our knowledge, studies where cyclodextrins impact not only the solubility but also permeability and, especially, distribution of the drugs are not numerous. We expect our efforts to increase such investigations would not be wasted.

In the present study the following aims were stated: to reveal the influence of the drug chemical structure on the solubility, distribution and permeability of the parent substances (iproniazid (IPN), isoniazid (INZ) and isonicotinamide (iNCT)), and to disclose how different cyclodextrins modulate the propensity of the model compound iproniazid to distribute between the organic and aqueous phases in the two-phase systems of the immiscible solvents and to penetrate the biomimetic lipophilic membrane. To this end, the solubility in 1-octanol and n-hexane, distribution in the 1-octanol/water and n-hexane/water systems and the permeability coefficients through the PermeaPad barrier were determined at different concentrations of two cyclodextrins: 2-hydroxypropyl-β-cyclodextrin and methylated β-cyclodextrin. From the obtained experimental results, the stability constants of drug–cyclodextrin complexes were evaluated by the phase-distribution method [21,28]. The consistency between the partition and permeation results was assessed.

## 2. Materials and Methods

### 2.1. Materials

The structures of the investigated compounds are depicted in Figure 2.

Information about the substances used in the experiments is represented in the Sample Table (Table 1).

Bidistilled water (2.1 μS cm^−1^ electrical conductivity) was used to prepare the buffer solution. Phosphate buffer pH 7.4 (*I* = 0.26 mol∙L^−1^) was prepared from KH_2_PO_4_ (9.1 g in 1 L) and Na_2_HPO_4_∙12H_2_O (23.6 g in 1 L) salts. The pH values of the solutions were monitored using a FG2-Kit pH meter (Mettler Toledo, Greifenzee, Switzerland) standardized with pH 4.00 and 7.00 solutions.

### 2.2. Solubility Determination in 1-Octanol and n-Hexane

The solubility of the investigated drugs in 1-octanol and n-hexane was determined by the shake-flask technique. To this end, the glass vials containing the solutions with the excess amount of the compound were placed in an air thermostat and stirred at a predetermined temperature for no less than 24 h (equilibrium state). After this, the suspensions were left for 5 h or more to precipitate the undissolved drug and centrifuged (Biofuge pico, Thermo Electron LED GmbH, Langenselbold, Germany) at a predetermined temperature during 20 min (10,000 rpm). The aliquots of the clear solutions were taken and examined via a spectrophotometer (Cary-50 Varian, Palo Alto, CA, USA, Software Version 3.00 (339)). The molar concentrations of the compounds in the solvents were calculated using calibration curves at the appropriate wavelength characteristic for each substance. The experimental results were reported as an average of at least three replicated experiments with an accuracy of 2–4%. In order to evaluate the thermodynamic parameters of the IPN dissolution in 1-octanol and n-hexane, the molar solubility ($S_{2}$) was recalculated to the mole fraction ($X_{2}$) by the equation:(1)$$X_{2}=\frac{M_{2}S_{2}}{S_{2}(M_{1}-M_{2})+1000\rho_{}},$$
where *M*_1_ and *M*_2_ are the molar masses of IPN and organic solvent, respectively, and *ρ* (g∙cm^−3^) is the density of the pure solvents. The linear functions describing the temperature dependences of the solubility within the temperature interval from 293.2 ± 0.5 K to 313.2 ± 0.5 K for both organic solvents were derived:(2)$$\ln X_{2}=A+\frac{B}{T}$$

The apparent solution enthalpies $\Delta H_{sol}^{0}$ were determined by the van’t Hoff equation:(3)$$\partial (\ln X_{2})/\partial T=\Delta H_{sol}^{0}/RT^{2}$$

The apparent dissolution Gibbs energies ($\Delta G_{sol}^{}$) and entropies ($\Delta S_{sol}^{}$) were calculated at a standard temperature of 298.2 K and 310.2 K:(4)$$\Delta G_{sol}^{298.2/310.2}=-RT(\ln X_{2}^{298.2/310.2})$$
(5)$$\Delta G_{sol}^{298.2/310.2}=\Delta H_{sol}^{0}-T\Delta S_{sol}^{298.2/310.2}$$
where *R* is the universal gas constant.

### 2.3. Distribution Examination in the Absence and Presence of Cyclodextrins

Apparent distribution coefficients of the compounds were measured in two systems: 1-octanol/buffer pH 7.4 and n-hexane/buffer pH 7.4 by the shake-flask technique [29] at 310.2 ± 0.5 K in the absence (for IPN, INZ and iNAM) and in the presence of cyclodextrins (HP-β-CD and M-β-CD) in the aqueous phase (for IPN only). The concentrations of cyclodextrins were: 0.0115 M, 0.025 M and 0.035 M. The experiments were carried out according to the literature [21,30]. In brief, the following experimental procedure was performed. Aqueous buffer pH 7.4 and organic solvent (1-octanol or n-hexane) were mixed during 48 h to obtain mutually saturated phases. The stock solutions containing the drug were prepared in buffer pH 7.4 saturated with the organic solvent. Both aliquots of the aqueous stock solution and organic solvent were placed in screwed glass vials and mixed at 310.2 ± 0.5 K during 3 days. The concentrations of the stock solutions were approximately 6.0 × 10^−3^ M for both systems. The volumes of the phases were 3:3 in the 1-octanol/buffer pH 7.4 systems and 6:3 and 10:3 in the n-hexane/buffer pH 7.4 systems, without and with cyclodextrins, respectively. After mixing, the equilibrium was achieved; the vials were maintained at 310.2 ± 0.5 K, and the phases were separated from each other. The drug concentrations were determined using spectrophotometer (Cary-50 Varian, Palo Alto, CA, USA, Software Version 3.00 (339)). The experiments were carried out at least four-fold with an accuracy of 2–4%. The distribution coefficients were calculated using the following equation:(6)$$D_{app}^{Org/buf}=\frac{C_{2}^{Org/buf}\cdot V^{buf/Org}}{C_{2}^{buf/Org}\cdot V^{Org/buf}}$$
where $D_{app}^{Org/buf}$ is the distribution coefficient calculated from the concentrations of the compounds expressed in molar scale, $C_{2}^{Org/buf}$ and $C_{2}^{buf/Org}$ are the molar concentrations of the drug in the mutually saturated organic solvent/buffer and buffer/organic solvent phases, respectively, and $V^{buf/Org}$ and $V^{Org/buf}$ are the volumes of the saturated buffer and organic solvent phases, respectively. In addition, the ΔlogD parameters were estimated taking into account the distribution coefficients both in the 1-octanol/buffer pH 7.4 and n-hexane/buffer H 7.4 systems. As Abraham et al. [31] stated, the difference between the distribution parameters in these systems (ΔlogD) can be applied to assess the specific interactions including a hydrogen-bonding impact to the drug transport properties and can be calculated as follows:(7)$$\Delta \log D=\log D_{app}^{oct/buf}-\log D_{app}^{hex/buf}$$

The association constants (*K_C_*) of IPN with cyclodextrins were determined by a partition coefficient (phase-distribution) method. Possible partition of CDs to the 1-octanol or n-hexane phases was assumed to be negligible since both cyclodextrins used in this study have high aqueous solubility (>600 mg∙mL^−1^) [32]. The *K_C_* values of the complexes between the drug and each CD were calculated assuming that in the presence of CD the distribution coefficient ($D_{app}^{Org/(buf+CD)}$) is defined as:(8)$$D_{app}^{Org/(buf+CD)}=\frac{C_{2}^{Org}}{C_{2}^{buf}(drug)+C_{2}^{buf}(drug•CD)}$$
where $C_{2}^{buf}(drug)$ and $C_{2}^{buf}(drug•CD)$ are the concentrations of the drug and drug complex with CD in the aqueous phase. The following equation was applied:(9)$$\frac{D_{app}^{Org/(buf+CD)}}{D_{app}^{Org/buf}}=1+K_{C}\cdot C_{CD}$$
where $D_{app}^{Org/buf}$ and $D_{app}^{Org/(buf+CD)}$ are the distribution coefficients of a compound in the absence and in the presence of CD in the aqueous phase, respectively. Similarly to the phase-solubility method of the stability constants determination, but bearing in mind an opposite trends in solubility and organic solvent/water partition (the higher the solubility, the lower the distribution coefficient), the following dependence was plotted to obtain the association constant (*K_C_*) and the stoichiometric ratio of the complex (*α*):(10)$$\log(\frac{D_{app}^{Org/buf}-D_{app}^{Org/(buf+CD)}}{D_{app}^{Org/buf}})=\log(K_{C})+\alpha \cdot \log(C_{CD})$$
where *C_CD_* is the concentration of cyclodextrin in the aqueous phase of the distribution system and the y-intercept produces log(*K_C_*).

### 2.4. Membrane Permeability Assay in the Absence and Presence of Cyclodextrins

Membrane permeability coefficients of the studied drugs were measured in the Franz diffusion cell of vertical type (PermeGear, Inc., Hellertown, PA, USA) with 7 mL/1 mL volumes of the donor and acceptor solutions through the reverse dialysis set up [33]. The PermeaPad barrier (PHABIOC, Fritz-Souchon-Str.27 32339 Espelkamp Germany, www.innome.de) (0.785 cm^2^ effective surface area) was mounted between the donor and acceptor chambers of the diffusion cell. The temperature was maintained at 310.2 ± 0.5 K throughout the experiment. Samples of 0.5 mL were withdrawn each 30 min from the acceptor solution and replaced with an equal amount of fresh buffer. Permeability experiments lasted over five hours in all cases. The solutions concentrations were analyzed via spectrophotometer (Spectramax 190; Molecular Devices Corporation, San Jose, CA, USA) in 96-well UV black plates (Costar) at a wavelength of 264 nm and 266 nm for IPN and INZ/iNAM, respectively. The dependences of the amount of the permeated drug over the surface area (d*Q*/*A*) on time (*t*) were plotted and flux (*J*) across the membrane was calculated by the equation:(11)$$J=\frac{dQ}{A\times dt}$$
The apparent permeability coefficients (*P_app_*) were determined by normalizing the flux measured over the concentration of the drug in the donor compartment (*C*_0_) by the equation:(12)$$P_{app}=\frac{J}{C_{0}}$$
Each permeability experiment was carried out in triplicate. The experiments were performed under sink conditions; that is, the drug concentration in the acceptor chamber did not exceed 10% of the drug concentration in the donor chamber at any time.

Similarly to the distribution, the permeability coefficients were obtained in the absence (for IPN, and iNAM) and in the presence of cyclodextrins (HP-β-CD and M-β-CD) (for IPN only) in the donor solution. The concentrations of cyclodextrins were 0.0115 M, 0.025 M and 0.035 M. The permeability coefficient for INZ through the PermeaPad barrier was determined in our previous investigation [11] and taken in the present study for the sake of comparison.

## 3. Results and Discussion

### 3.1. Assessment of the Solubility, Distribution and Permeability for IPN, INZ and iNAM at 310.2 K

The molar solubility values of the studied structurally parent substances were determined in 1-octanol and n-hexane at 310.2 ± 0.5 K (Table 2). The solubility was shown to be 338-fold, 130-fold and 115-fold higher in 1-octanol than in n-hexane for IPN, INZ and iNAM, respectively. A higher solubility in 1-octanol is explained by its ability to form the specific interactions with the solvent, as compared to n-hexane—an inert solvent—which interacts with a solute only non-specifically. Similar trends in the solubility variations between the compounds in both organic solvents were revealed: S_2_ (IPN) > S_2_ (INZ) > S_2_ (iNAM), in spite of the rather different natures of 1-octanol and n-hexane. An essentially greater value of S_2_ determined for IPN (in comparison with both INZ and iNAM) is most probably due to the presence of two methyl groups in the structure of the amide substituent. The electron-donor CH_3_ group with a positive induction effect acts as a hydrophobic substituent, increasing the lipophilicity of a molecule in the whole, thus facilitating the dissolution in organic solvents (especially, in highly lipophilic 1-octanol).

In order to evaluate the driving forces of the solvation processes in the pharmacologically significant solvents—1-octanol and n-hexane—usually used for the evaluation of the drug lipophilicity and hydrogen bonding potential, the solubility of IPN designated as a model compound was studied at five temperatures (Table 2).

As follows from Table 2, in both solvents the solubility increases with the temperature growth. The temperature dependences of the mole fraction solubility are illustrated in Appendix A and used further to calculate the thermodynamic functions of the dissolution processes (Table 3) at a standard temperature of 298.2 K and 310.2 K (used as the temperature very close to a healthy human).

The following conclusions can be derived from Table 3. The dissolution process is more favorable in 1-octanol as compared to n-hexane. In both solvents it is determined by the enthalpy, but the difference between the enthalpy and entropy contributions is not essential (especially in 1-octanol). The greater positive enthalpy term is a driving force of the unfavorable dissolution processes in these two solvents. A more than two-fold difference $\Delta H_{sol}^{}$ > $T\Delta S_{sol}^{}$ exists in n-hexane, and a favorable positive entropy value is not enough to overlap the enthalpy one. From the dissolution thermodynamic functions, the hypothetical transfer n-hexane→1-octanol simulating the blood-brain penetration can be disclosed. As follows from Table 3, the transferring is favorable (negative $\Delta G_{tr}^{}$ value), in that the lower temperature facilitates the process. The hypothetical transition from n-hexane to 1-octanol is driven and determined by the negative enthalpy (~80%). A small negative entropy contribution—$T\Delta S_{tr}^{}$ indicates some ordering of the system upon the movement of the IPN molecules from n-hexane to 1-octanol. At the next step it seemed interesting to compare the obtained thermodynamic results for IPN with those reported in our earlier study [34] on the INZ solubility in 1-octanol at 298.2 K ($\Delta G_{sol}^{298.2}$ = 13.9 kJ∙mol^−1^, $\Delta H_{sol}^{298.2}$ = 24.6 kJ∙mol^−1^, $T\Delta S_{sol}^{298.2}$ = 10.7 kJ∙mol^−1^). The comparative analysis demonstrated that both an increase in $\Delta H_{sol}^{298.2}$ by 4.1 kJ∙mol^−1^ and a decrease in $T\Delta S_{sol}^{298.2}$ by 3.6 kJ∙mol^−1^ produce a greater value of $\Delta G_{sol}^{298.2}$, reducing the solubility of INZ in this solvent.

For a long time, lipophilicity expressed as the distribution coefficient in the 1-octanol/water system (log$D_{}^{oct/water}$) has served as the major determinant of drugs dissolution and membrane permeation [35]. The range of lipophilicity (log$D_{}^{oct/water}$ = 1–3) indicates the optimal gastrointestinal absorption after per oral administration from a good balance of solubility and passive diffusion permeability [36]. Besides this, the ∆log*D* parameter of Seiler [31] is also indicative to evaluate the blood–brain penetration and hydrogen bonding potential of drug.

From the results of the distribution experiments (Table 4, Figure 3a) carried out for three structurally parent substances—IPN, INZ and iNAM—we assessed their lipophilicity and hydrogen bonding potential from log$D_{app}^{oct/buf}$ and log$D_{app}^{hex/buf}$ according to Equation (7). Importantly, the positive values of log$D_{app}^{Org/buf}$ are indicative of shifting the equilibrium to the organic phase (higher drug concentration in the organic phase), whereas the negative ones show the compound to exist predominantly in the aqueous phase. Consequently, only in the case of IPN distribution in the 1-octanol/buffer system the equilibrium is shifted to the organic phase. The experimental concentrations of the compounds in the organic and aqueous phases of the distribution systems are given in Appendix A.

Notably, the literature value of log$D_{\exp}^{oct/pH7.4}$ for IPN at 293.2 K (20 °C) [37] was shown to be 0.35 ± 0.10, which is in agreement with log$D_{\exp}^{oct/pH7.4}$ = 0.423 ± 0.015 derived in the present study at 310.2 K. Expectedly, the descending order of the compounds distribution coefficients was revealed to be the same as for their solubility in the studied solvents (Table 4, Figure 3a): log$D_{app}^{Org/buf}$(IPN) > log$D_{app}^{Org/buf}$(INZ) > log$D_{app}^{Org/buf}$(iNAM). The values of ∆log*D* are similar for INZ and iNAM and higher by 0.45 units for IPN, demonstrating higher propensity of this compound to specific interactions (hydrogen bonding). Notably, according to the pK_a_ values calculated by the pDISOL-X program [38] (*pK*_a1_ = 3.62, 3.80 and 3.60 and *pK_a_*_2_ = 11.28, 13.61 and 13.71 for IPN, INZ and iNAM, respectively) all the compounds exist in the unionized form in the aqueous phase, which evidences that the ionization state does not change upon the partitioning between the water buffer and organic solvent.

Membrane permeability along with solubility in biological fluids serves as an important determinant of drug bioavailability. Moreover, the information on the dissolution/distribution/permeability properties represents the physicochemical profile which facilitates the search for the perspective drug candidates in a homologous series of chemical entities [39]. Therefore, at the next step of the investigation we assessed another key parameter determining the bioavailability—permeability through the model lipophilic membrane (PermeaPad)—for all three substances. The results, presented as the donor solution concentrations, steady-state fluxes and permeability coefficients, are given in Appendix A. For the sake of comparison, the P_app_ values for the compounds are illustrated as a diagram in Figure 3b next to the distribution coefficients. Similarly to the distribution coefficients, the permeability coefficients are reduced in the range: IPN (2.03·10^−5^ cm·s^−1^) > INZ (1.44·10^−5^ cm·s^−1^) > iNAM (0.73·10^−5^ cm·s^−1^). According to the general rule, the more bulky IPN molecule can be expected to diffuse slower than the other two compounds. However, it is not so. Most probably, higher lipophilicity of IPN coming from the presence of the methyl groups promotes the interactions of the compound with the components of the phospholipid PermeaPad barrier, thus facilitating the transition through the membrane.

In order to disclose the trends in structure–property relationships on the example of the studied substances, we calculated the HYBOT physicochemical descriptors (primarily the molecular polarizability, donor and acceptor ability to hydrogen bonds formation) by the program package HYBOT-PLUS (version of 2003) in Windows [40] and correlated them to the solubility, distribution parameters and permeability coefficients. Fully recognizing inconsistence of the correlations based on three points only, we identified the trends rather than correlations. The seeking with the HYBOT descriptors has led to the relationships between some of the considered parameters and the polarizability descriptor. Interestingly, as far back as 2003, the importance of polarizability in chemical and biological interactions was proved by Hansh et al. [41]. In our study, we derived the linear logarithmic dependences connecting the solubility in 1-octanol and n-hexane with the distribution coefficients in the 1-octanol/buffer and n-hexane/buffer systems. The respective plots and equations are given in Figure 4a–d.

As it is seen from Figure 4, the correlations with polarizability are better for the solubility and distribution in 1-octanol than those in n-hexane. The review of numerous investigations [42,43] showed that the growth of the molecule size and polarizability, as well as the increase in lipophilicity reduce aqueous solubility, increase the solubility in 1-octanol and shift the distribution to the organic phase. This regularity was proved on the example of the studied substances. The obtained correlation dependences seem to be useful for further design, screening and investigations of new biologically active compounds belonging to the pyridinecarboxamide derivatives.

### 3.2. Distribution and Permeability of IPN in the Presence of HP-β-CD and M-β-CD in Aqueous Phase

In this section we evaluated how the presence of CDs impacts the distribution and permeability of a model substance, IPN. The values of the distribution coefficients in the presence of cyclodextrins are given in Table 4. Appendix A collects the raw experimental data on the concentrations of IPN in the organic and water phases of the distribution systems. As follows from Table 4, the distribution coefficients of IPN decrease upon growing the cyclodextrin concentration in the aqueous phase of both studied distribution systems, but this process is more pronounced in the case of HP-β-CD. For a better visualization, the distribution coefficients dependences are illustrated in Figure 5. Since the negative slopes of the plots in Figure 5a,b are indicative of decreasing the drug concentration in the organic phase and increasing in the aqueous one, it becomes evident that the transferring of the compound from water to the organic phase is inhibited more intensively by HP-β-CD as compared to M-β-CD. To disclose the possible reasons of this phenomenon, Figure 5c illustrates the variations of the ∆log*D* parameter upon the growing of CD concentration.

As follows from Figure 5c, the hydrogen bonding (specific forces) contribution to the distribution/lipophilicity of IPN (∆log*D*) is greater at higher amounts of CDs in the aqueous phases, especially in the case of HP-β-CD for which a more pronounced growth of the slope is observed. This growth is responsible of a more intensive decreasing of $D_{app}^{oct/pH7.4}$ upon the HP-β-CD concentration growth (as compared to M-β-CD) in accordance with the well-established rule that for a better transition to the lipophilic medium the hydrogen bonding should be minimized [31]. In turn, a more intensive interaction of the compound with HP-β-CD is expected since the number of the hydrogen bond donors and acceptors is greater in the HP-β-CD molecule (as compared to M-β-CD). As a result, a stronger specific interaction of IPN with HP-β-CD in aqueous phase hampers the transition of the compound to the lipophilic 1-octanol to a greater extent than M-β-CD presented in the aqueous phase. In addition, higher lipophilicity of the methylated cyclodextrin [44] (compared to HP-β-CD) facilitates a weaker interaction with the hydrophilic substance (IPN), thus also hampering its transferring from water to 1-octanol.

To approve the validity of the distribution results interpretation we used the distribution coefficients at different CDs concentrations for the determination of the association constants of IPN with the studied cyclodextrins. As shown by Másson et al. [21] for hydrophilic drugs (a case of IPN), this method can be successfully used for the simultaneous determination of the association constant with cyclodextrins and partition coefficient. The association constants (*K_C_*) of IPN with HP-β-CD and M-β-CD were determined according to the phase-distribution method using Equations (8)–(10). To this end, the plots of the dependences (Equation (10)) were constructed and are illustrated in Figure 6.

The values of the association constants derived from the dependences in Figure 6 are tabulated along with the parameters of the correlation equations (Table 5).

As follows from the values of α parameters, the stoichiometry of all the complexes is 1:1. According to the association constants, IPN interacts strongly with HP-β-CD due to the presence of the peripheral hydroxyl groups in its structure. The complexes of IPN with both cyclodextrins can be classified as extremely weak [45]. Markedly, the values of the association constants derived from the distribution experiments in the 1-octanol/buffer pH 7.4 system are not so far from those obtained using the n-hexane/buffer pH 7.4 partitioning. Considering the correlation coefficients and Fisher criteria (Table 5), both systems seem to be valid for the evaluation of the complex stability. In so doing, the correlation parameters are better, firstly, in the case of the 1-octanol containing systems as compared to n-hexane ones, and, secondly, with HP-β-CD as compared to M-β-CD. The results from the association constants calculations support the conclusions made from the comparative analysis of the distribution coefficients in the presence of HP-β-CD and M-β-CD. A somewhat lower value of the association constant of the IPN/M-β-CD complex (as compared to IPN/HP-β-CD) is the reason for a less pronounced decrease of the distribution coefficients ($D_{app}^{oct/pH7.4}$ and $D_{app}^{hex/pH7.4}$) in the presence of an elevating concentration of CD in the aqueous phase of the partition system.

The permeability coefficients (*P_app_*) of IPN through the PermeaPad barrier in the presence of different concentrations of cyclodextrins are presented in Appendix A. For better visualization, the values of *P_app_* are illustrated as a diagram in Figure 7. Importantly, the PermeaPad barrier is lipophilic in nature, forms the bilayer structure and is composed of soy phosphatidylcholine (S-100) containing choline, glycerol, glycolipids triglycerides and fatty acids, which make it very close to the biological membranes. It is clear that permeability of a solute depends on the additives in the examined solution (cyclodextrins in this study) which influence the solvation/re-solvation processes and interactions upon the drug passing through the membrane. On the other hand, the lipophilic/hydrophilic character of the solute is also of a paramount importance.

Appendix A and Figure 7 demonstrate different trends in the permeability variations of IPN in the presence of HP-β-CD and M-β-CD. A slight decrease of the permeability coefficient values (up to 1.78 × 10^−5^ cm∙s^−1^) upon growing the HP-β-CD concentration is observed, possibly as a result of the reduction of the free (non-complexed) IPN molecules concentration in the presence of HP-β-CD in the donor solution. Besides this, HP-β-CD was shown to have a potential to protect the liposomal membrane from damage via replacing the water molecules at the liposome surface due to numerous hydrogen-bond donors and acceptors in the structure [46] which can inhibit the diffusion of IPN through the liposomal PermeaPad barrier. In contrast, the presence of M-β-CD enhances (up to 2.87 × 10^−5^ cm∙s^−1^) the diffusion rate of IPN through the PermeaPad barrier. As shown in the literature, cyclodextrins can interact with the membrane components [47] and, thus, modulate the permeability. M-β-CD was shown to act as a permeation enhancer [44]. Mura et al. [48] demonstrated a two-fold enhancement of the permeability coefficient of clonazepam in the presence of methylated cyclodextrin and attributed this effect to the interaction with the bilayer of the liposomal vehicles of the artificial membrane. Most likely, the mechanism of the permeability improvement is based on the capability of M-β-CD to extract the lipid components from the PermeaPad barrier, to complex them, thus disrupting the ordered lipid structure of the barrier and reducing its tightness in a random way. All the above arguments explain the opposed trends in the permeability of IPN through the PermeaPad barrier in the presence of HP-β-CD and M-β-CD.

In the case of HP-β-CD, the IPN permeation regularity matches the distribution behavior which seems to be reliable since 1-octanol serves as a model of the phospholipids of biological membranes [49]. The same tendency was obtained in our earlier study on the enzalutamide and apalutamide [25].

## 4. Conclusions

The experiments on the solubility (1-octanol and n-hexane), distribution (1-octanol/buffer pH 7.4 and n-hexane/buffer pH 7.4 systems) and permeation (PermeaPad barrier) of three parent compounds containing the pyridinecarboxamide scaffold (iproniazid (IPN), isoniazid (INZ) and isonicotinamide (iNCT)) were carried out at 310.2 K. As a result, the following regularities have been disclosed. The similar trends in the values of the solubility, distribution and permeability coefficients were revealed for studied substances: IPN > INZ > iNAM. The plots of the solubility and distribution coefficients on the Hydrogen Bond Thermodynamics (HYBOT) descriptors provided the linear dependences with the molecular polarizability. The obtained data proved that the growth of the molecule size and polarizability results in increasing the solubility in 1-octanol and n-hexane, as well as the distribution coefficients in the 1-octanol/buffer pH 7.4 and n-hexane/buffer pH 7.4 systems. The equations derived from the dependences are expected to be useful for further design of the drug formulations based of the compounds belonging to the pyridinecarboxamide class.

The impact of cyclodextrins (HP-β-CD and M-β-CD) on the distribution and permeability behavior of the model compound (IPN) was examined using different concentrations of cyclodextrins in the aqueous phases of the partitioning systems and in the solutions for the diffusion experiments. The distribution coefficients of IPN decreased upon growing the cyclodextrin concentration in the aqueous phase of both studied distribution systems, but this process was shown to be more pronounced in the case of HP-β-CD. This was attributed to the enhancing of the specific interaction of IPN with HP-β-CD in the aqueous phase, greater lipophilicity of the methylated cyclodextrin and higher contribution of the hydrogen bonding (specific forces) to the distribution/lipophilicity of IPN in the case of HP-β-CD. The distribution coefficients at different CDs concentrations were used for the determination of the association constants of IPN with the studied cyclodextrins. A stronger interaction of IPN with HP-β-CD was estimated according to the association constants most probably due to the presence of the peripheral hydroxyl groups in this cyclodextrin. Different trends in the permeability variations in the presence of HP-β-CD and M-β-CD were revealed as a result of possible interactions of the last cyclodextrin with the components of the phospholipid membrane (PermeaPad barrier). Therefore, M-β-CD can be considered as a permeation enhancer for IPN. We expected the results obtained in the presence of cyclodextrins could be applied for the modulation of the properties of pharmaceutical preparations based on pyridinecarboxamide drugs and cyclodextrins.

## Figures and Tables

**Figure 1 pharmaceutics-15-01531-f001:**
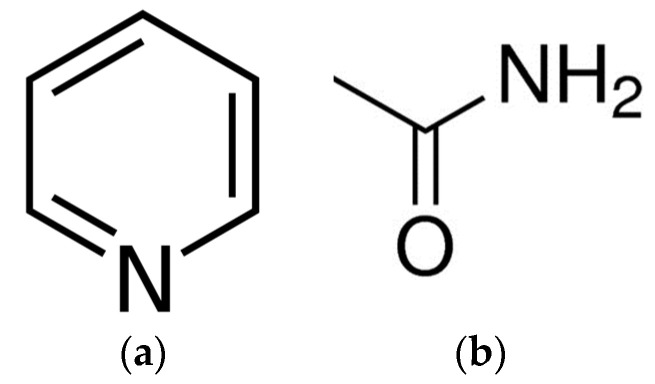
The structures of pyridine fragment (**a**), and carboxamide group (**b**).

**Figure 2 pharmaceutics-15-01531-f002:**
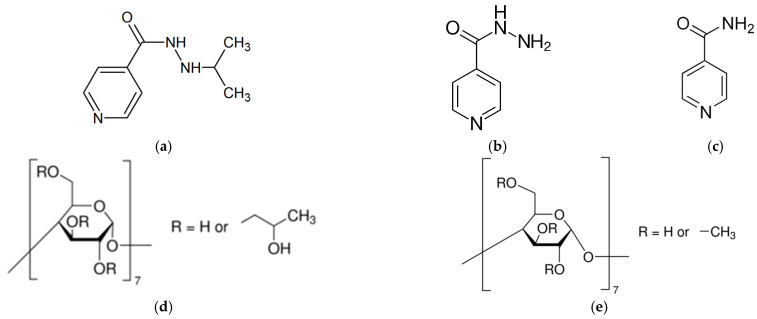
Iproniazid (IPN) (**a**), isoniazid (INZ) (**b**), isonicotinamide (iNCT) (**c**), 2-hydroxypropyl-β-cyclodextrin (HP-β-CD) (**d**), and methylated β-cyclodextrin (M-β-CD) (**e**) structures.

**Figure 3 pharmaceutics-15-01531-f003:**
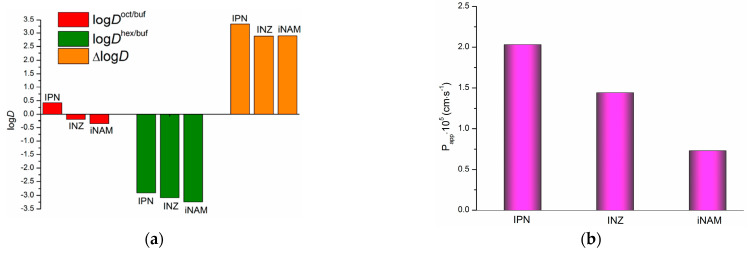
Distribution coefficients of IPN, INZ, and iNAM in the 1-octanol/buffer pH 7.4, n-hexane/buffer pH 7.4 systems and ∆log*D* parameter (log scale) (**a**); and permeability coefficients through the PermeaPad barrier (**b**) at 310.2 K.

**Figure 4 pharmaceutics-15-01531-f004:**
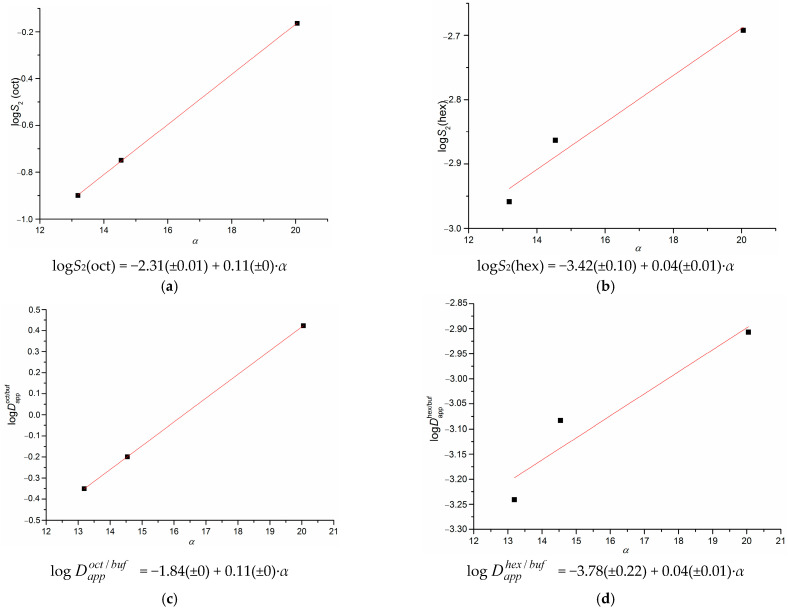
Dependences of the parameters obtained in the present study: solubility in 1-octanol (**a**), solubility in n-hexane (**b**), distribution coefficient in the 1-octanol/buffer pH 7.4 system (**c**), distribution coefficient in the n-hexane/buffer pH 7.4 system (**d**).

**Figure 5 pharmaceutics-15-01531-f005:**
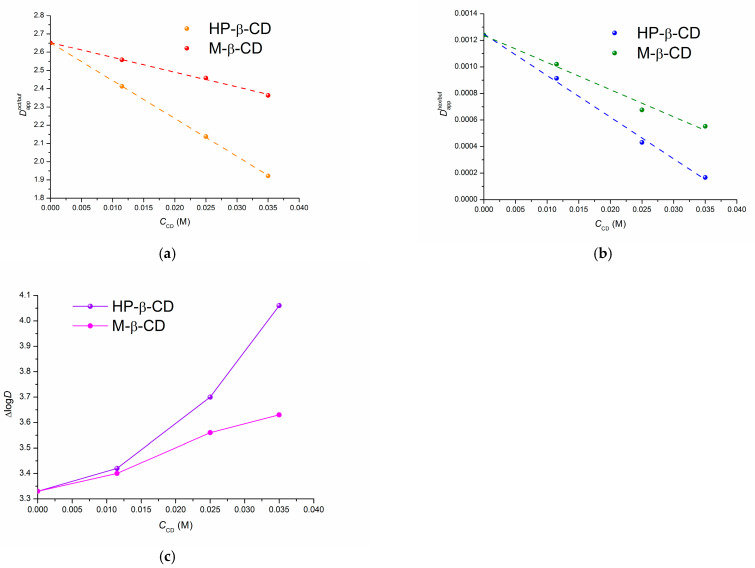
Dependences of the distribution coefficients on the cyclodextrin concentration in the aqueous phase: (**a**) 1-octanol/buffer pH 7.4 system, (**b**) n-hexane/buffer pH 7.4 system, (**c**) ∆log*D* parameter, 310.2 K.

**Figure 6 pharmaceutics-15-01531-f006:**
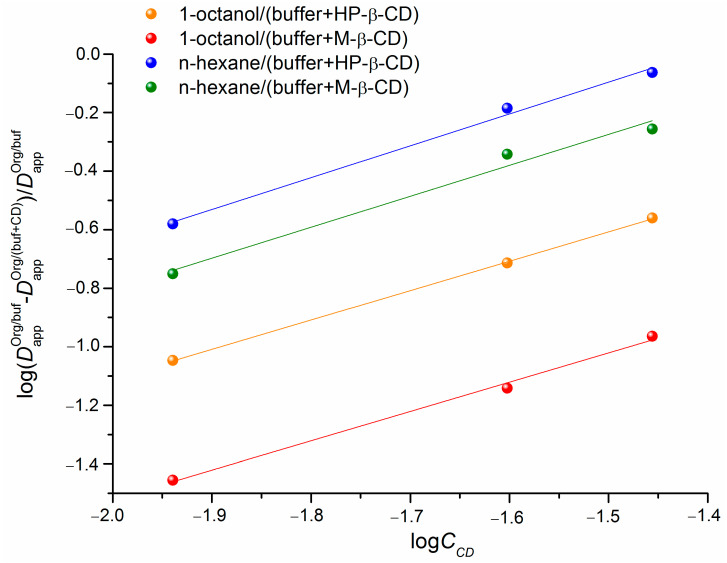
The plots of the logarithmic dependences of $(D_{app}^{Org/buf}-D_{app}^{Org/(buf+CD)})/D_{app}^{Org/buf}$ on the cyclodextrin concentration $C_{CD}$ from the distribution coefficients in the 1-octanol/buffer pH 7.4 and n-hexane/buffer pH 7.4 systems measured at 310.2 K.

**Figure 7 pharmaceutics-15-01531-f007:**
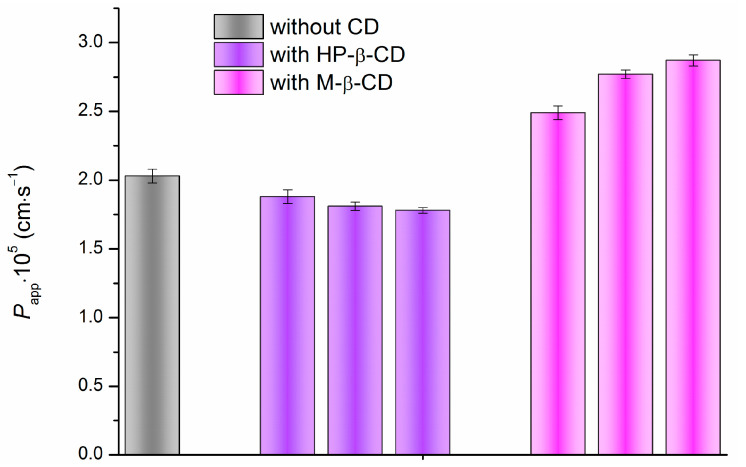
Permeability coefficients (*P_app_*) of IPN without and with different concentrations of cyclodextrins in the solutions measured in the Franz diffusion cell through the PermeaPad barrier at 310.2 K. The columns are placed in the ascending order of CD concentrations, from left to right: 0.015 M, 0.025 M, 0.035 M.

**Table 1 pharmaceutics-15-01531-t001:** Sample Table.

Compound Name	CAS Register No.	Source	Mass Fraction Purity	Purification Method
Iproniazid (N′-isopropyl isonicotinic hydrazide)	54-92-2	Weng Jiang Reagent Co.	0.96	none
Isoniazid (isonicotinic acid hydrazide)	54-85-3	Acros organics	≥0.99	none
Isonicotinamide (pyridine-4-carboxylic acid amide)	1453-82-3	Acros organics	0.99	none
2-Hydroxypropyl-β-cyclodextrin	128446-35-5	Sigma-Aldrich	≥0.96	none
Methylated β-cyclodextrin		Aldrich	-	none
1-octanol	111–87-5	Sigma-Aldrich	≥0.99	none
n-hexane	110–54-3	Sigma-Aldrich	≥0.99	none
Potassium dihydrogen phosphate	7778–77-0	Merck	≥0.99	none
Disodium hydrogen phosphate dodecahydrate	10039–32-4	Merck	≥0.99	none

**Table 2 pharmaceutics-15-01531-t002:** Molar solubility (*S*_2_) of IPN, INZ and iNAM in 1-octanol and n-hexane.

Compound	T (K)	1-Octanol	n-Hexane
		*S_2_* × 10^1^ (M)	*S_2_* × 10^3^ (M)
IPN	290.2	4.09 ± 0.08	0.72 ± 0.01
IPN	293.2	4.44 ± 0.09	0.83 ± 0.03
IPN	298.2	5.06 ± 0.06	1.07 ± 0.02
IPN	303.2	5.73 ± 0.11	1.42 ± 0.03
IPN	310.2	6.86 ± 0.18	2.03 ± 0.04
INZ	310.2	^a^ 1.78 ± 0.04	1.37 ± 0.05
iNAM	310.2	1.26 ± 0.02	1.10 ± 0.04

^a^ taken from [34]. Standard uncertainty *u*(*T*) = 0.5 K; Relative standard uncertainties *u_r_*(*S*) = 0.04.

**Table 3 pharmaceutics-15-01531-t003:** Thermodynamic functions (kJ·mol^−1^) of IPN solubility in n-hexane and 1-octanol and of the hypothetical n-hexane → 1-octanol transferring calculated at 298.2 K and 310.2 K.

Solvent	$\Delta G_{sol}^{298.2}$	$\Delta G_{sol}^{310.2}$	$\Delta H_{sol}^{298.2/310.2}$	$T\Delta S_{sol}^{298.2}$	$T\Delta S_{sol}^{310.2}$	${\;}^{3}\zeta_{Hsol}^{}$
^1^ 1-octanol	6.2 ± 0.1	5.6 ± 0.1	20.5 ± 0.2	14.3 ± 0.4	14.9 ± 0.4	58.9/57.9
^2^ n-hexane	22.0 ± 0.4	21.2 ± 0.4	40.0 ± 0.9	18.0 ± 0.8	18.8 ± 0.8	69.0/68.0
n-hexane → 1-octanol	$\Delta G_{tr}^{298.2}$	$\Delta G_{tr}^{310.2}$	$\Delta H_{tr}^{298.2/310.2}$	$T\Delta S_{tr}^{298.2}$	$T\Delta S_{tr}^{310.2}$	^4^ $\zeta_{Htr}^{}$
	−15.8	−15.6	−19.5	−3.7	−3.9	84.1/83.3

^1^ ln*X*_2_ = (5.79 ± 0.08) − (2469 ± 23)/*T*; *r* = 0.9999; *σ* = 5.07·10^−5^; *n* = 5; ^2^ ln*X*_2_ = (7.28 ± 0.35) − (4809 ± 104)/*T*; *r* = 0.9993; *σ* = 1.03·10^−3^; *n* = 5; ^3^
$\zeta_{Hsol}^{}$ = (|$\Delta H_{sol}^{}$|/(|$\Delta H_{sol}^{}$| + |$T\Delta S_{sol}^{}$|))·100%; ^4^
$\zeta_{Htr}^{}$ = (|$\Delta H_{tr}^{}$|/(|$\Delta H_{tr}^{}$| + |$T\Delta S_{tr}^{}$|))·100%.

**Table 4 pharmaceutics-15-01531-t004:** Distribution coefficients $D_{app}^{Org/buf}$ (log$D_{app}^{Org/buf}$) of IPN, INZ and iNAM in the 1-octanol/buffer pH 7.4 and n-hexane/buffer pH 7.4 systems and for IPN in the presence of HP-β-CD and M-β-CD in buffer phases at 310.2 K.

System	$D_{app}^{oct/buff}$ $\;(\log D_{app}^{oct/buff})$	$D_{app}^{hex/buff}$ $\cdot 10^{4}\;(\log D_{app}^{hex/buff})$	Δlog*D*
IPN	2.651 ± 0.093 (0.423)	12.400 ± 0.191 (−2.907)	3.33
INZ	0.633 ± 0.021 (−0.199)	8.254 ± 0.188 (−3.083)	2.88
iNAM	0.446 ± 0.013 (−0.351)	5.742 ± 0.172 (−3.241)	2.89
IPN/0.0115 M HP-β-CD	2.413 ± 0.086 (0.383)	9.138 ± 0.161 (−3.039)	3.42
IPN/0.025 M HP-β-CD	2.138 ± 0.087 (0.330)	4.311 ± 0.113 (−3.365)	3.70
IPN/0.035 M HP-β-CD	1.922 ± 0.096 (0.284)	1.672 ± 0.061 (−3.777)	4.06
IPN/0.0115 M M-β-CD	2.558 ± 0.081 (0.408)	10.200 ± 0.210 (−2.991)	3.40
IPN/0.025 M M-β-CD	2.459 ± 0.068 (0.391)	6.763 ± 0.122 (−3.170)	3.56
IPN/0.035 M M-β-CD	2.363 ± 0.072 (0.373)	5.525 ± 0.128 (−3.258)	3.63

**Table 5 pharmaceutics-15-01531-t005:** Association constants (*K_C_*) of IPN complexes with cyclodextrins from the distribution experiments in the 1-octanol/buffer pH 7.4 and n-hexane/buffer pH 7.4 systems at 310.2 K.

Cyclodextrin	*K_C_*	^a^ *α*	^b^ *R*	^c^ *σ*	^d^ *F*
		1-Octanol/Buffer pH 7.4 System			
HP-β-CD	0.899 ± 0.022	1.004 ± 0.014	0.9999	2.28·10^−5^	5441.26
M-β-CD	0.479 ± 0.109	1.000 ± 0.065	0.9979	5.22·10^−4^	235.47
		**n-hexane/buffer pH 7.4 system**			
HP-β-CD	1.536 ± 0.128	1.087 ± 0.076	0.9975	7.22·10^−4^	201.42
M-β-CD	1.311 ± 0.241	1.057 ± 0.143	0.9909	2.56·10^−3^	53.97

^a^—the parameter indicating the complex stoichiometry; ^b^—the pair correlation coefficient; ^c^—the residual sum of squares; ^d^—the Fisher criterion.

## Data Availability

No data was used for the research described in the article.

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
