# Peer review of "Modulation of Distribution and Diffusion through the Lipophilic Membrane with Cyclodextrins Exemplified by a Model Pyridinecarboxamide Derivative"

_pharmaceutics, 2023, doi:10.3390/pharmaceutics15051531_

Round 1

Reviewer 1 Report

This manuscript demonstrates solution thermodynamic and membrane flux parameters determined across three different pyridine derivatives as a model systes.  The main goal of this study was to elucidate relationships between the structures of these compounds and physicochemical characteristics. The authors provide well executed and well designed experiments to determine these parameters.  Some of the results such as solubility trends and organic/water partition coefficeint trends were obvious based on structures.  The authors also investigate equilibria with cyclodextrins to determine association constants and concentration impacts on partition coefficients between two different CD molcules.  Several conclusions are drawn from the work in this manuscript concerning the impacts of hydrogen bonding on solubility and partitioning, and the impacts of these interactions on complexation with CD.  The appraoches and results obtained are proposed to be useful for the design of pharmaceutical formulations.

Specific comments:

1.    The main conclusion in the absact and elaborated upon in the results and discussion are that the structure of the CD enhances permeability of the model compound IPN.   Little discussion is provided as to the mechanism of this enhamcement.  Based on the association constants determined, the mechanism is unlikely to orignate from solubility enhancement by the M-b-CD through complexation, for example by replenishing free drug at the unstirred boundary layer.  Can the authors provide a mechanistic interpretation for why M-b-CD enhances the permeability coefficient in a concentration dependent manner?  For example, is the assumption that M-b-CD does not partition into the membrane correct, or are its impacts directly from interactions on the donor side of the Franz cell?  Can it somehow act as a transmembrane shuttle, how?

2.    It is understood that the use of 310.15 K is used based on the exact conversion of zero Celius to 273.15 K, and this convention is widely used throughout the thermodynamic literature.  This convention is not used in pharmaceutical sciences and reported temperature values are usually reported to be reflective of the level of temperature control exerted by the experimental apparatus.  Most experimental apparatus available cannot hold a solution at temperature with an accuracy of of 0.05 K (i.e. the last significant figure used throughout the manuscript).  For example, if an oven controls to +/- 0.5K, then temperature cannot be reported to a accuracy narrower than this range. Can the authors include a statement on the accuracy and precision of temperature control in their experimental apparatus so that the reader can better understand the quality of temperature control used in these experiments?   

3.    The solution thermodynamics assessment is interesting, but it is unclear how these values relate to the conclusions of the paper.  For example, can the authors instruct how would a drug formulator might use these values to mechanistically interpret solubility data?  Is there an insight that these values can provide that relates to a QSAR relationship?

4.    Line 268, possible typo "The range of lipophilicity (log = 1÷3)"

5.    Line 268, drug formulators generally recognize the Lipinski "rule of 5", where a combination of logP<5 among other variables is often associated with reasonable bioavailability of a drug. This rule is often violated, but is a good general rule of thumb.  Why is the more narrow range of 1-3 stated as indicative of good gastrointestinal absorption?

6.    A literature value for logD(oct/pH 7.4) is given in the range of 0.35 +/-0.1 and this is stated to be significantly different form the value of 0.423 determined in this work.  Are these values significantly different?

7.    Several key figures called out in the text are provided as Supporting information.  I would suggest Figure S2 be provided in the main text as it is the topic of a length discussion and a conclusion is directly drawn from it on page 9. 

8.    Please use a grammatical editor on this manuscript as multiple errors were noted throughout. 

Author Response

Response to Review_1

This manuscript demonstrates solution thermodynamic and membrane flux parameters determined across three different pyridine derivatives as a model systes.  The main goal of this study was to elucidate relationships between the structures of these compounds and physicochemical characteristics. The authors provide well executed and well designed experiments to determine these parameters.  Some of the results such as solubility trends and organic/water partition coefficeint trends were obvious based on structures.  The authors also investigate equilibria with cyclodextrins to determine association constants and concentration impacts on partition coefficients between two different CD molcules.  Several conclusions are drawn from the work in this manuscript concerning the impacts of hydrogen bonding on solubility and partitioning, and the impacts of these interactions on complexation with CD.  The appraoches and results obtained are proposed to be useful for the design of pharmaceutical formulations.

Reply:

 Dear Reviewer, thank you very much for careful consideration of our manuscript. We've taken into account all your comments and suggestions. The corrections are marked in red in the manuscript body.

Specific comments:

Comment:

  1. The main conclusion in the absact and elaborated upon in the results and discussion are that the structure of the CD enhances permeability of the model compound IPN.   Little discussion is provided as to the mechanism of this enhamcement.  Based on the association constants determined, the mechanism is unlikely to orignate from solubility enhancement by the M-b-CD through complexation, for example by replenishing free drug at the unstirred boundary layer.  Can the authors provide a mechanistic interpretation for why M-b-CD enhances the permeability coefficient in a concentration dependent manner?  For example, is the assumption that M-b-CD does not partition into the membrane correct, or are its impacts directly from interactions on the donor side of the Franz cell?  Can it somehow act as a transmembrane shuttle, how?

Reply:

Most likely, the mechanism of the permeability improvement is based on the capability of M-β-CD to extract the lipid components from the PermeaPad barrier, complex them, thus disrupting the ordered lipid structure of the barrier and reducing its tightness in a random way.

Short explanation of the permeability enhancement of IPN by M-b-CD has been provided in the manuscript (Section 3.2., Page 14, after Figure 7). 

Comment:

  1. It is understood that the use of 310.15 K is used based on the exact conversion of zero Celius to 273.15 K, and this convention is widely used throughout the thermodynamic literature.  This convention is not used in pharmaceutical sciences and reported temperature values are usually reported to be reflective of the level of temperature control exerted by the experimental apparatus.  Most experimental apparatus available cannot hold a solution at temperature with an accuracy of of 0.05 K (i.e. the last significant figure used throughout the manuscript).  For example, if an oven controls to +/- 0.5K, then temperature cannot be reported to a accuracy narrower than this range. Can the authors include a statement on the accuracy and precision of temperature control in their experimental apparatus so that the reader can better understand the quality of temperature control used in these experiments? 

 Reply:

Thank you very much for useful recommendation. We reconsidered the accuracy of the temperature control throughout the experiments in accordance with the experimental apparatus. T=310.2±0.5 K is the precision of temperature control. The corrections have been done throughout the manuscript and Supporting Information files.

Comment:

  1. The solution thermodynamics assessment is interesting, but it is unclear how these values relate to the conclusions of the paper.  For example, can the authors instruct how would a drug formulator might use these values to mechanistically interpret solubility data?  Is there an insight that these values can provide that relates to a QSAR relationship?

Reply:

In the present study the solution thermodynamics was evaluated for IPN in 1-octanol and n-hexane. This information seems to be useful since 1-octanol and n-hexane are usually used for the evaluation of the drug lipophilicity (log), brain penetration (log) and hydrogen bonding potential of the substance (∆logD). In particular, drug lipophilicity is of a paramount significance because it often serves as a descriptor for drug bioavailability, bioactivity and, as follows, relate to a QSAR. Importantly, in our study the linear dependences of both the solubility in 1-octanol and the logparameter on the structural descriptor - polarizability - were obtained based on the structurally related substances IPN, INZ, and iNAM. In addition, similar dependences were obtained for the solubility and distribution coefficient with n-hexane (but the correlation coefficient value was lower as compared to 1-octanol).

The respective information has been highlighted in the Conclusions Section (Page 15) according to the Reviewer recommendations.

Comment:

  1. Line 268, possible typo "The range of lipophilicity (log = 1÷3)"

Reply:

The sentence (Page 8) has been corrected.

Comment:

  1. Line 268, drug formulators generally recognize the Lipinski "rule of 5", where a combination of logP<5 among other variables is often associated with reasonable bioavailability of a drug. This rule is often violated, but is a good general rule of thumb.  Why is the more narrow range of 1-3 stated as indicative of good gastrointestinal absorption?

Reply:

Besides the Lipinski "rule of 5", logP<5, the range of lipophilicity logD7.4=1-3 was stated by Kerns and Di (Kerns, E.H.; Di, L. Druglike properties: Concepts, structure design and methods, 2008; pp. 44–45.) as the ideal range for good intestinal absorption, owing to a good balance of solubility and passive diffusion permeability. The authors underlined that in the case of logD7.4=3-5, the compounds have good permeability but absorption is lower, owing to lower solubility. In the manuscript the range of the ideal lipophilicity is supported by the reference (Page 8).

Comment:

  1. A literature value for logD(oct/pH 7.4) is given in the range of 0.35 +/-0.1 and this is stated to be significantly different form the value of 0.423 determined in this work.  Are these values significantly different?

Reply:

"the literature value of log  for IPN at 293.15 K (20 oC) [Gulyaeva et al. Eur. J. Med. Chem. 2003, 38, 391–396] was shown to be 0.35±0.10 that is in agreement with log=0.423 derived at the same temperature in the present study."  These values are in consistence. The sentence has been corrected (Page 9).

Comment:

  1. Several key figures called out in the text are provided as Supporting information.  I would suggest Figure S2 be provided in the main text as it is the topic of a length discussion and a conclusion is directly drawn from it on page 9.

 Reply:

Figure S2 and Table S1 have been moved to the main text as Figure 4 and Table 3. The rest Figures and Tables have been re-numbered, respectively.

Comments on the Quality of English Language

Comment:

  1. Please use a grammatical editor on this manuscript as multiple errors were noted throughout. 

Reply:

The manuscript has been checked for the quality of English Language

Reviewer 2 Report

The  article Modulation of distribution and diffusion through the lipophilic membrane with cyclodextrins exemplified by a model pyridine amide derivative is well-written, but I have some comments: 

Please put the 13, 14, and 15 references in square brackets (in the introduction part).

The structures of the investigated compounds must moved in the Introduction part, not in the Materials part.

At point 2.4 please specify how long samples were taken. Specify the concentration of the drug in the analyzed samples. Specify the specific wavelengths for each compound.

The information: ˮPermeaPad barrier was purchased from the PHABIOC shop (Fritz-Souchon-Str.27 112 32339 Espelkamp Germany) (www.innome.de).ˮ presented at point 2.1. also appears at point 2.4. Please deleted from point 2.1. Is not necessary.

You give in the Results part the permeability coefficients (Papp) of IPN through the PermeaPad barrier in the presence of different concentrations of cyclodextrins. Please also provide information about the permeability of the other 2 compounds: isoniazid (INZ) and  isonicotinamide (iNCT).

I have no comments

Author Response

Response to Review_2

Dear Reviewer, thank you very much for careful consideration of our manuscript. We've taken into account all your comments and suggestions. The corrections are marked in red in the manuscript body.

Comment:

Please put the 13, 14, and 15 references in square brackets (in the introduction part).

Reply:

References 13, 14, and 15 have been put in square brackets.

Comment:

The structures of the investigated compounds must moved in the Introduction part, not in the Materials part.

Reply:

The structures of the investigated compounds have been moved to the Introduction part.

Comment:

At point 2.4 please specify how long samples were taken. Specify the concentration of the drug in the analyzed samples. Specify the specific wavelengths for each compound.

Reply:

The following specific experimental details have been introduced in Section 2.4 of Methods Section.   Permeability experiments lasted over five hours in all cases. The concentrations of the drug in the analyzed samples (along with the Fluxes and Permeability coefficients) are given in Table S2 (SI file). The specific wavelengths for each compound (IPN: λmax=264 nm; INZ: λmax=266 nm; iNAM: λmax=266 nm;) have been indicated.

Comment:

The information: ˮPermeaPad barrier was purchased from the PHABIOC shop (Fritz-Souchon-Str.27 112 32339 Espelkamp Germany) (www.innome.de).ˮ presented at point 2.1. also appears at point 2.4. Please deleted from point 2.1. Is not necessary.

Reply:

The information on the PermeaPad barrier has been deleted from Section 2.1.

Comment:

You give in the Results part the permeability coefficients (Papp) of IPN through the PermeaPad barrier in the presence of different concentrations of cyclodextrins. Please also provide information about the permeability of the other 2 compounds: isoniazid (INZ) and  isonicotinamide (iNCT).

Reply:

The permeability of isoniazid (INZ) and  isonicotinamide (iNCT) have been introduced in the Results (Section 3.1). In this section the comparative analysis of the permeability coefficients for IPN, INZ, and iNAM is presented.

The impact of cyclodextrins on the distribution and permeability was studied only on the example of IPN selected as a model compound.

Reviewer 3 Report

The authors present their study on the effect of cyclodextrins on the behavior of pyridine amide derivatives.  I would suggest the authors consider the following:

1.  I believe the better descriptor for these compounds would be pyridinecarboxamides rather than pyridine amides.

2.  Some relevant structures should be provided earlier in the introduction so the reader understands the general structure of the compound discussed. 

3.  Table S1 is discussed in detail on page 7.  This Table should probably be included in the main text.

4.  I'm unclear what is meant by the equation: log=1 "divided by" 3 on line 268.

5.  It would be helpful when discussing the graphs to provide a statement for each that says, for example "positive values mean X and negative values mean Y" so the reader doesn't have to figure this out on their own.  For example, I had to go back to the equation for distribution coefficients to understand the trends observed in Figure 3.  If the authors stated "the negative slope of the line indicates decreasing concentrations in the organic phase and increasing concentrations in the aqueous phase", it would have been easier to understand more quickly.

6.  It is still not clear to me why HP-b-CD decreases the permeability coefficient relative to IPN without cyclodextrin.  Perhaps this could be addressed better.

With attention to the questions above this paper may warrant publication.

English is OK but could use some improvement if accepted.

Author Response

Response to Review_3

Dear Reviewer, thank you very much for careful consideration of our manuscript. We've taken into account all your comments and suggestions. The corrections are marked in red in the manuscript body.

Comment:

  1. I believe the better descriptor for these compounds would be pyridinecarboxamides rather than pyridine amides.

Reply:

Thank you for useful suggestion. We replaced " pyridine amides " with " pyridinecarboxamides " throughout the manuscript body.

Comment:

  1. Some relevant structures should be provided earlier in the introduction so the reader understands the general structure of the compound discussed. 

Reply:

To facilitate the understanding of the general structure of the discussed compounds, additionally Figure 1 (Page 2) has been inserted in the introduction in order to provide the relevant structures of the pyridine and carboxamide fragments. The rest Figures have been re-numbered, respectively.

Comment:

  1. Table S1 is discussed in detail on page 7.  This Table should probably be included in the main text.

Reply:

Table S1 has been moved to the main text as Table 3 (Page 7). The rest Tables have been re-numbered, respectively.

Comment:

  1. I'm unclear what is meant by the equation: log=1 "divided by" 3 on line 268.

Reply:

The correct expression of the range of lipophilicity for the optimal gastrointestinal absorption has been introduced as: log =1‒3 (Page 8).

Comment:

  1. It would be helpful when discussing the graphs to provide a statement for each that says, for example "positive values mean X and negative values mean Y" so the reader doesn't have to figure this out on their own.  For example, I had to go back to the equation for distribution coefficients to understand the trends observed in Figure 3.  If the authors stated "the negative slope of the line indicates decreasing concentrations in the organic phase and increasing concentrations in the aqueous phase", it would have been easier to understand more quickly.

Reply:

According to the Reviewer suggestion, in order to make the discussion of the plots more clear to the reader, we have added the plot explanations concerning Figure 3 (Distribution coefficients of IPN, INZ, and iNAM in the 1-octanol/buffer pH 7.4, n-hexane/buffer pH 7.4 systems, and ∆logD parameter (log scale) (a); and permeability coefficients through the PermeaPad barrier (b) at 310.15 K.) (Page 8), and Figure 5 (Dependences of the distribution coefficients on the cyclodextrin concentration in the aqueous phase: (a) 1-octanol/buffer pH 7.4 system, (b) n-hexane/buffer pH 7.4 system, (c) ∆logD parameter, 310.15 K.) (Page 11).

Comment:

  1. It is still not clear to me why HP-b-CD decreases the permeability coefficient relative to IPN without cyclodextrin.  Perhaps this could be addressed better.

Reply:

The process of drug passage through the membrane is a complicated phenomenon, especially if the components of the "real" biological barriers are in its content (PermeaPad meets the case).

In the case of HP-b-CD one of the reasons of the permeability decrease is the reduction of the free (non-complexed) IPN molecules concentration in the presence of HP-b-CD in the donor solution. (We agree with the Reviewer that this point was missed in the text.) One can see that this decrease is not essential in accordance with very low drug/HP-b-CD association constant. In addition, as it was indicated in the text, "HP-β-CD was shown to have a potential to protect the liposomal membrane from damage via replacing the water molecules at the liposome surface due to numerous hydrogen-bond donors and acceptors in the structure". Most probably, these two factors facilitate the decrease of the IPN permeability in the presence of HP-b-CD.

According to the Reviewer recommendation, we expanded the explanation of the HP-b-CD impact on the permeability reduction. The issue of the reduction of the free (non-complexed) IPN molecules concentration in the presence of HP-b-CD has been elucidated. (Page 14).

We checked and improved English throughout the manuscript.

Round 2

Reviewer 3 Report

Thank you to the authors for addressing my concerns.  The paper is publishable at this time.